# Determinants of Quality of Life in the COVID-19 Pandemic Situation among Persons Using Psychological Help at Various Stages of the Pandemic

**DOI:** 10.3390/ijerph19106023

**Published:** 2022-05-16

**Authors:** Joanna Chwaszcz, Michał Wiechetek, Rafał P. Bartczuk, Iwona Niewiadomska, Patrycja Wośko

**Affiliations:** Institute of Psychology, The John Paul II Catholic University of Lublin, 20-950 Lublin, Poland; michal.wiechetek@kul.pl (M.W.); bartczuk@kul.pl (R.P.B.); iwona.niewiadomska@kul.pl (I.N.); patrycja.wosko@kul.pl (P.W.)

**Keywords:** COVID-19, coronavirus, quality of life, coping, resources, well-being, mental health

## Abstract

This article presents the results of three surveys conducted during the initial stage of the COVID-19 pandemic, in March 2020 and in late June/early July 2020, when pandemic restrictions were in force. The surveys covered patients who had used psychological assistance before the pandemic. two were cross-sectional and one was longitudinal. The first survey involved 270 people (age: M = 29.59, SD = 10.74, women 79.3%), and the second one covered 117 subjects (age: M = 29.40, SD = 11.49, women 85.5%). The third, longitudinal, survey covered 83 subjects (age: M = 26.61, SD = 7.17, women 89.2%). In our research we used the Conservation of Resources Evaluation questionnaire, the abbreviated version of the Coping Orientation to Problems Experienced, the WHO Quality of Life Scale, and a questionnaire for collecting sociodemographic information. Our analysis of the quality-of-life correlates in the two cross-sectional studies leads to an observation that for people using psychological help, the constant determinants of quality of life during a pandemic are high gain in resources and little loss of resources. In the first phase of the pandemic, active strategies were not linked to the sense of quality of life. This sense, however, was diminished by a number of negative strategies, such as denial, venting, substance use, restraint, and self-blame. After 3 months of the ongoing pandemic, perceived quality of life was positively correlated with strategies related to seeking instrumental support and active coping. this most likely points to a process of adapting to a difficult situation. The results of our longitudinal surveys demonstrate increasing escapism. Our attempt at explaining which factors determined the quality of life after 3 months of the ongoing pandemic showed that the crucial factors are: a sense of quality of life before the occurrence of the pandemic, changes in the distribution of resilience-oriented resources, changed frequency of using passive strategies, and active ways of coping—but only after 3 months into the pandemic. The results thus obtained can be used both in prevention and in work with persons affected by the negative consequences of the COVID-19 pandemic.

## 1. Introduction

The pandemic time has required new models of adaptation. United Nations (UN) experts underscore that during the coronavirus disease 2019 (COVID-19) pandemic, many people experienced anxiety caused by social isolation, fear of infection, family members getting infected, and their loss. At the same time a huge many have lost or face the loss of their jobs and therefore their livelihoods. Additionally, false information about the virus and vaccinations is being circulated, thus aggravating people’s uncertainty about the future [1,2]. In a UN report on the connections between COVID-19 and public health, it was pointed out that people may turn to various methods of coping with stressors that can harm their mental health [3]. In psychiatry, the term “pandemic acute stress disorder” has emerged [4]. The most typical and common feature of pandemic acute stress disorder is a prolonged anxiety reaction and inability to disengage oneself from the continuing experience of trauma. One observes prolonged fear and a sense of helplessness and, if escaping is impossible, spells of panic, despair, and a feeling of hopelessness. The symptoms classified in pandemic acute stress disorder fall into individual categories of obsessions, depressed mood, dissociative symptoms, avoidance, and excessive arousal. In practice, diagnosing disorders resulting from the experience of COVID-19 has only begun. The psychological consequences of living during the pandemic have a varying intensity, but they affect entire populations [4].

### 1.1. Quality of Life and Its Determinants

The self-assessed measure defining people’s functioning in the pandemic situation is quality of life (QoL), related to personal adaptation and satisfaction with a life under specific conditions. Life circumstances can be divided into structures forming one’s external capital and processes occurring between the individual and the environment. These processes can be interpreted as gains in and losses of resources both external and internal. On the one hand, an individual shapes their environment, but on the other, they adapt to environmental demands, gaining balance in doing so. Disturbing this balance gives rise to unpleasant emotions, elevated stress levels, and resource loss. The spread of the coronavirus caused schools, companies, and public areas to be closed all around the world and forced many communities to impose bans on leaving home, thereby causing stress in people of all ages, genders, and socioeconomic statuses [5]. However, humans strive for balance by adapting to ever-changing demands of the environment. This process takes a toll on one’s resources, which the individual forfeits in a difficult situation and then tries to regain, or the individual tries to gain new ones that are equally attractive and essential for coping with the changing circumstances. Loss of resources is associated with increasing stress levels. Gaining resources reduces stress and enhances QoL [6]. Among the factors determining the effectiveness of this process is the presence of resources in the social space surrounding the person. This presence has been severely curtailed during the pandemic. Research involving 1500 individuals from 47 countries showed that perceived QoL is connected with physical health, spirituality, emotional loneliness, social loneliness, self-efficacy, and a sense of job security [7]. During the second pandemic wave, research conducted with three cultural groups of 694 participants in Israel showed significant relationships between emotional suffering and resources such as intrapersonal hope and a sense of coherence [8]. Research conducted among 1186 middle-aged adults in Turkey showed that significant predictors of life satisfaction during the pandemic include the sense of life and hope [9]. Research results from a study conducted during the pandemic in Italy of a sample of 2784 participants aged 18 to 85 years confirmed a significant relationship between subjective resources and perceived mental well-being. People with a low sense of coherence experienced significantly lower quality of life in the mental domain. Moreover, coherence plays an essential role in shaping the relationship between disease experiences and mental well-being [10]. During the COVID-19 pandemic, these constructs have been impaired due to the disease itself, fear of it, and all the restrictions imposed on community life and ways of working, which contributed to reducing perceived QoL. In a Brazilian study with a sample of 426 participants (7 to 80 years old), it was proven that all aspects of QoL were significantly reduced in both sexes and among all age groups, except for children; stress levels increased significantly during the pandemic [11].

Another determinant of QoL are the strategies a person uses to cope with stress. A new, unknown situation of stress, which the COVID-19 pandemic certainly is, calls for novel coping strategies. Coping with stress during the pandemic can be a key indicator of adaptation. Choosing strategies and adapting them to the existing circumstances has a bearing on, among other things, the possibility of gaining resources [12,13,14,15]. Active strategies are linked to resource gain and a better QoL [16]. Long-term studies conducted in Germany from March to May 2020 on a sample of 93 persons demonstrated that coping associated with commitment, such as problem solving, was positively correlated with well-being. The opposite was true for escape and blame coping [17]. The results of research conducted with 521 Chinese adults proved that active strategies such as hygiene practices, seeking support, and positive reframing were positively linked to hope [18]. Another study conducted in the UK in May 2020, which covered 555 subjects, indicated that avoidance strategies were positively correlated with all indicators of distress and negatively with well-being [19]. In a study of a Brazilian population, the use of an avoidance strategy was linked to an increased likelihood of severe depression, whereas active coping and reframing were associated with mental health [20]. In a study carried out in Turkey with a population of 4624 adults, it was demonstrated that coping with COVID-19 was a mediating link between anxiety linked with COVID-19 and general health [21]. Research on 577 Polish students from 17 universities showed that during the coronavirus pandemic, they were most likely to employ coping strategies such as acceptance, planning, and seeking emotional support. The least likely strategies were substance use, denial, behavioural withdrawal, and religious coping. The most significant strategies related to good adaptation in the pandemic time were seeking instrumental support, seeking emotional support, and planning [22]. Cross-sectional research carried out in Australia with a sample of 1495 subjects led to the conclusion that the coping strategies that positively correlated with mental health were psychological reframing, acceptance, and humour. Conversely, self-blame, venting, behavioural withdrawal, and diversion were associated with poorer mental health [23]. Another study, covering 11,227 subjects from 30 countries on all continents, proved that meaning-focused coping was the strongest positive predictor of physical and mental health of all the coping strategies [24]. Because of the wide variety of strategies used to cope with the stress related to the COVID-19 pandemic, it seems that the way we feel is affected not as much by the event itself but rather by our psychological processing of the event. An analysis of coping should consider individuals in their social contexts and their ongoing involvement in stressful situations [25].

### 1.2. Development of the Pandemic and the Restrictions It Entails

The present research was conducted in two periods: (1) in March 2020, when the pandemic was erupting, and (2) in late June/early July 2020, when the restrictions of the first wave were being lifted. A description of the COVID-19 pandemic situation in Poland at that time is essential for a full understanding of the survey results.

During the first survey (March 2020), the first case of SARS-CoV-2 infection in Poland was reported. In response to that, the Polish government introduced the first restrictions, for example, on visiting hospital patients; a sanitation regime was also imposed. On 10 March, the government started to implement changes regarding public gatherings: All mass events were cancelled, and within the next few days, the number of worshippers at church services was limited to 50 and then to 5. Next, schools, colleges, kindergartens, and crèches were closed. On 14 March, a state of epidemic threat was declared in Poland. The daily number of infections grew to about 5 thousand, and the mortality rate was up to 180 per day. During our second survey (May/June 2020), Polish citizens were obliged to cover their mouths and noses, a quarantine was imposed on individuals who had been in contact with infected persons, and schools and colleges continued to teach remotely; classroom teaching of practical subjects was permitted, but most educational facilities did not take advantage of this, and catering outlets were closed. There was a limit on the number of passengers using public transport and on church attendance. From 18 May, beauty parlours and hairdressers started to operate under the sanitation regime. From 30 May, people engaging in outdoor sports were no longer required to wear masks. Sports event audiences of up to 250 participants were allowed. The limit on the number of customers inside shops, street markets, or post offices was lifted. From 1 June, domestic passenger flights were restored. The daily number of identified infections was between 19 and 25 thousand, and the daily mortality rate ranged from 900 to 1137 people. As of 13 June 2020, Poland restored unrestricted border traffic across the internal borders of the EU. This meant that travellers did not have to undergo quarantine. On 22 June, Poland restored international railway traffic across the internal borders of the EU. Restrictions on the operation of shopping centres were lifted.

### 1.3. Aim

The aim of this survey was to examine the QoL of people living in the pandemic and its correlates initially and after 3 months. We predicted that QoL and the related factors would evolve during the pandemic due to imposed and removed restrictions on social activity. This is, first and foremost, connected with limited access to resources and the unchanging situation despite active strategies undertaken to cope with stress. Another aim was to explore the determinants of people’s perceived QoL 3 months into the pandemic. An insight into factors determining perceived QoL during a time of pandemic will make it possible to understand mechanisms affecting QoL in stressful situations and under resource deprivation.

## 2. Materials and Methods

### 2.1. Subjects and Procedure

This paper presents three studies, two cross-sectional and one longitudinal. The two cross-sectional studies involved persons who had at least once used psychological help such as psychological support, consultancy, or counselling due to mild psychological problems offered by nongovernmental organizations (NGOs) in Lublin (eastern Poland) in the 12 months preceding the pandemic outbreak. Additional inclusion criteria were: age over 18 years and inclusion in the register of clients of a counselling centre run by an NGO. The surveys were conducted remotely via the Google platform, and an invitation to the study was sent via e-mail. The respondents were requested to fill out a questionnaire and informed that participation in the survey was voluntary. The information thus obtained would serve as the basis of an international discussion of the COVID-19 experience. The research took place in March and late June/early July 2020. The first survey involved 270 people, and the second covered 117 subjects. In all three studies, participants completed the same questionnaire. Completing the survey took about 20 min. There were different subjects in each study. The research was conducted at different moments of the pandemic because we wanted to capture the adaptation process to a problematic situation in the first phase and after the initial impacts had passed.

### 2.2. Ethics

The research was conducted in accordance with the ethical standards of a responsible committee on human experimentation (institutional or regional) and with the Declaration of Helsinki (the 2013 revision). To comply with the ethical standards, the research was conducted according to the standards of good research practice recommended by the American Psychological Association. The participants were informed about the confidentiality and anonymity of the research and that they had the right not to participate in the survey. Additionally, the very first question on the survey was: Do you agree to voluntarily participate in the study? The possible answers were yes and no. No personal data were collected.

### 2.3. Research Instruments

The surveys were part of a larger project. In this article, we used the results of the Conservation of Resources Evaluation (COR-E), the modified Brief Coping Orientation to Problems Experienced questionnaire (Brief-COPE), the short version of the WHO Quality of Life Questionnaire (WHOQL-BREF), and a questionnaire for collecting sociodemographic data.

#### 2.3.1. Gain in and Loss of Resources

Distribution of resources was measured with the COR-E questionnaire [6,26]. The questionnaire contains a list of 74 resources. Our study utilized the Polish adaptation by Chwaszcz and Bartczuk [27]. The participants responded to individual items by choosing their answers on a 5-point scale (from 1 = *Not at all* to 5 = *To a great degree*) in two categories: gain and loss. In order to analyse the COR-E structure, a bifactorial model was used, based on which we identified a global resource factor and 7 group factors [27]: (1) *management resources* (F1), indispensable for managing one’s life and essential for managing other resources such as sense of control over one’s life, acting as a leader, and organisational skills; (2) *social status resources* (F2), encompassing aspects like financial stability, satisfying earnings, adequate status at work, and savings or money to use in unforeseen situations; (3) *resilience resources* (F3), encompassing personal qualities and conditions that allow people to function optimally and increase their resilience: family stability, sense of emotional closeness, vitality/stamina, sense of self-worth, hopefulness, sense of self-efficacy, sense of achievement, self-pride, optimism, etc.; (4) *family resources* (F4), encompassing aspects like a good marriage, good relationships with one’s children, helping to look after children, healthy children, healthy spouse/partner, feeling of closeness to spouse/partner, and provision of essential resources for children; (5) *material status resources* (F5), including the following: necessary household equipment, a home larger than necessary, a home that meets one’s needs, suitable clothing, more clothes than necessary, and proper furnishing of the home. These resources include the aspects of consumer culture that determine one’s material social status; (6) *growth resources* (F6), including money for personal development, membership in organisations where people can share their interests, and involvement in church life/a religious community. This factor refers to the human need for self-fulfilment. It also encompasses development of competences, knowledge, and skills which can be bought with money (e.g., a postgraduate course of study, vocational course, etc.); development of one’s interests (e.g., in a chess club); interpersonal, social, or religious development; and (7) *community resources* (F7), covering the health of family members, loved ones, and friends; the loyalty of friends, having at least one friend, company, etc. Community resources are related to relationships with loved ones, concern for loved ones, functioning in a group which achieves common goals, etc. In the research discussed in this paper, Cronbach’s alpha for the reliability of individual dimensions of the COR-E was acceptable for all dimensions and measurements. In the first round of the cross-sectional research, it ranged from 0.69 (gain in growth resources) to 0.99 (loss in resourcefulness). In the second round of the cross-sectional research, it ranged from 0.70 (gain in growth resources) to 0.98 (both gain and loss in resourcefulness). In the longitudinal survey, reliability ranged from 0.64 (loss in growth resources) to 0.99 (loss in resourcefulness) for the first measurement and from 0.63 (loss in growth resources) to 0.98 (loss in resourcefulness) for the second measurement.

#### 2.3.2. Coping Strategies

Coping strategies were measured using the modified Brief Coping Orientation to Problems Experienced (Brief-COPE) questionnaire by Carver [28], specifically the Polish adaptation by Juczyński and Ogińska-Bulik [29]. The method enables an assessment of how often an individual uses 14 different coping strategies: Active coping, Planning, Positive reframing, Acceptance, Humour, Religion, Seeking emotional support, Seeking instrumental support, Self-distraction, Denial, Venting, Psychoactive substance use, Restraint, Self-blame.

In the original Brief-COPE, respondents are asked how they usually cope with difficult situations and respond to each of 28 items on a 4-point scale from 0 = *I haven’t been doing this at all* to 3 = *I’ve been doing this a lot*. The results for individual strategies are calculated as the means of the items that they comprise. In this study, the instructions for the respondents were slightly modified to adapt to coping with the COVID-19 situation. The instructions read as follows: “People react differently when they encounter difficult or stressful events in their lives. The questions below are intended to determine how you respond to various experiences related to the current situation of the coronavirus (COVID-19) pandemic.” In the research discussed in this paper, Cronbach’s alpha of the reliability of individual dimensions of the Brief-COPE varied. In the first round of the cross-sectional survey, it ranged from 0.31 (Venting) to 0.95 (Psychoactive Substance Use). In the second round of the cross-sectional survey, it ranged from 0.41 (Self-Distraction) to 0.92 (Psychoactive Substance Use). In the longitudinal survey, reliability ranged from 0.41 (Self-Distraction) to 0.90 (Turning to Religion) for the first measurement and from 0.01 (Venting) to 0.92 (Turning to Religion) for the second, where reliability was the lowest. On five of the Brief-COPE dimensions, reliability was less than 0.60. Apart from Venting, the other dimensions were scored as follows: Active Coping (0.59), Acceptance (0.52), Humour (0.14), and Restraint (0.52). The reliability of the modified version of Brief-COPE was comparable with that of the Polish adaptation of the original scale.

#### 2.3.3. Quality of Life

Sense of QoL was measured with the World Health Organization Quality of Life-BREF (WHOQOL-BREF), an abbreviated generic QoL scale developed by the WHO [30,31]. It is an international, cross-culturally comparable QoL assessment instrument that was developed simultaneously with 15 international field centres. It assesses the individual’s perceptions in the context of their cultures and value systems and their personal goals, standards, and concerns. The WHOQOL-BREF comprises 26 items, of which 24 measure 4 broad QoL domains: physical health, psychological health, social relationships, and environment. In the study, a global QoL index was also computed that consisted of all the items forming individual dimensions. In the research presented in this paper, Cronbach’s alpha of the reliability of individual COR-E dimensions was acceptable for all dimensions and measurements. In the first round of the cross-sectional survey, it ranged from 0.70 (Physical Health) to 0.91 (Global Quality of Life). In the second round of the cross-sectional survey, it ranged from 0.78 (Physical Health) to 0.93 (Global Quality of Life). In the longitudinal survey, reliability ranged from 0.68 (Physical Health) to 0.90 (Global Quality of Life) for the first measurement and from 0.65 (Social Relations) to 0.89 (Global Quality of Life) for the second.

The following Cronbach’s alpha reliability coefficients were obtained: Physical Health, 0.70; Psychological Health, 0.82; Social Relationships, 0.73; and Environment, 0.78.

#### 2.3.4. Sociodemographic Variables

Our analysis at this stage relied on the questions used in the survey set concerning age, gender, education, and place of residence.

### 2.4. Statistical Analysis

Data analysis was carried out using SPSS ver. 27.0 [32]. Descriptive statistics were used during the analyses to present the intensity levels of the results for individual variables. Pearson’s correlation coefficient was used to estimate the relationships between variables in Surveys 1 and 2. In our analysis of the results of the third survey (longitudinal), we used a *t*-test for dependent samples in order to estimate severity variations in two periods of the pandemic. Additionally, we conducted a stepwise regression analysis to check which differences between the variables of the longitudinal survey were the best predictors of the overall QoL level. A threshold of *p* < 0.05 was assumed for statistical significance.

## 3. Results

Descriptive statistics and first-order correlations of the variables used in Surveys 1 and 2 are presented in Appendix A.

### 3.1. Study 1: Onset of the Pandemic

The first survey covered subjects aged 18 to 70 years (*M* = 29.59, *SD* = 10.74), with women accounting for 79.3% (*n* = 214). Respondents from urban centres of more than 50 thousand inhabitants accounted for 53.7% (*n* = 145), countryside dwellers constituted 27.8% (*n* = 75), and inhabitants of towns up to 50 thousand constituted 18.5% (*n* = 50) of the surveyed population. The subjects in this group had mainly secondary education (50.7%, *n* = 137) and higher (47.8%, *n* = 129). Primary and vocational education was represented by only 1.4% (*n* = 4) of the respondents.

During the onset of the pandemic, in March 2020, the global QoL perceived by the subjects was positively correlated with their gains in all categories of resources: management, social status, resilience, family, material status, growth, and community.

We also demonstrated in the first survey that the global QoL perceived by the subjects was correlated with their individual ways of coping. QoL increased along with the use of such strategies as: planning activities, attribution of positive meaning to an existing situation, and seeking emotional support. A lowering of QoL occurred if the following strategies were employed: denial of a difficult situation, venting unpleasant emotions, psychoactive substance use, restraint (passivity), and self-blame.

### 3.2. Study 2: 3 Months into the Pandemic

The second survey covered persons aged 19 to 60 years (*M* = 29.40, *SD* = 11.49), where women accounted for 85.5% (*n* = 100) of the subjects. Dominant in this group were subjects from cities of over 50 thousand inhabitants (50.4%, *n* = 59) and towns up to 50 thousand inhabitants (29.1%, *n* = 34). One fifth, 20.5% (*n* = 24), of the subjects were from rural areas. The educational structure of this group was dominated by secondary education (57.3%, *n* = 67) and higher education (41%, *n* = 48). Primary and vocational education was represented by only 1.8% (*n* = 2) of the respondents.

In the second survey, after 3 months since the onset of the pandemic (June/July 2020), just like when it began, perceived global QoL was positively linked to gains in all categories of resources and linked negatively with their losses. Three months into the pandemic, resource gains and losses continued to be factors affecting the subjects’ perceived global QoL.

After 3 months of the ongoing pandemic and living with socioeconomic restrictions, perceived global QoL was positively correlated with coping in difficult situations related to activity planning, positive reframing, and seeking emotional support, just as at the outbreak of the pandemic. Additionally, during this time, global QoL was positively related to active coping and seeking instrumental support. These strategies were not associated with the QoL experienced during the pandemic onset; rather, they were activated while the pandemic was ongoing. Consequently, it can be concluded that for the surveyed group, the factors minimizing perceived stress connected with the pandemic and discomfort related to the many restrictions were seeking instrumental support, i.e., practical advice/solutions, and active coping. In the first phase of the pandemic, these strategies were not effective, but after people adapted to the difficult situation, they turned out to be highly relevant (though weakly perceived) to QoL. Strategies that were negatively correlated with well-being were self-blame and restraint. After 3 months, strategies such as denying the existing situation and negative venting were not strongly linked to perceived QoL.

### 3.3. Study 3 (Longitudinal)

The longitudinal study involved 83 individuals. The survey had the same methodology as the one employed in the cross-sectional surveys. The subjects completed questionnaires in March and late June/early July 2020. Three people aged 22 to 50 years (*M* = 26.61, *SD* = 7.17) took part, with women prevailing at *n* = 74 (89.2%). The majority of the respondents came from cities larger than 50 thousand (*n* = 45, 54.2%). Towns of up to 50 thousand were represented by 24.1% (*n* = 20), whereas rural inhabitants accounted for 21.7% (*n* = 18). Persons in this group had mainly secondary education (62.7%, *n* = 52) and higher (37.3%, *n* = 31). No respondents had primary or vocational education.

The longitudinal survey conducted twice with the same group showed no changes in QoL between the two measurements during the COVID-19 pandemic (cf. Table 1). However, changes were found in the gain of resources and the coping strategies applied. In the case of stress coping strategies, differences were noted between the two surveys in three strategies: psychoactive substance use, restraint, and self-blame. For all the said strategies, a considerable increase in the frequency of their use was noted in the second survey. Regarding the resources, the differences concerned only gains and were related to: global resources, management, social status, resilience, and community resources. As the pandemic wore on, there was a decrease in resource gain in the said categories.

The next step was a regression analysis. Owing to the relatively small number of subjects participating in the longitudinal study and based on the results of our earlier surveys of a population in Poland, we decided to reduce the 14 dimensions of stress coping strategies to two higher-order factors: engaged coping (including active approach, seeking social support, and cognitive mastery of the situation) and escape coping (which included evasion countermeasures, denial, emotional response, and escape into psychoactive substances and religion) cf. [33]. In this longitudinal survey, Cronbach’s alphas for these dimensions of stress coping were, respectively, 0.77 and 0.66 in the first and second waves of the first survey and 0.74 and 0.56 in the first and second waves of the second.

A hierarchical multiple regression analysis was performed to assess the impacts of several variables measured in the first and second waves on QoL measured in the second wave. In the first step of the procedure, the initial QoL measurement was input as the baseline. In the second step, the gain and loss of resilience resources from the first measurement was added, as well as the engaged and escape coping styles from the first survey. In the third step, differences between the second and first longitudinal surveys were input with respect to variables associated with resilience resources (gain and loss) and composite stress coping strategies. Table 2 presents the results of regression analyses for QoL in the second measurement. The results show that good QoL predictors in the second measurement were: the initial QoL and the initial level of engaged coping style, changes in resource gain and loss, and changes related to escape strategies. Our analysis showed that a higher QoL was experienced by those who already had a higher QoL at the pandemic onset and who used engaged coping strategies. Further into the pandemic, the level of loss decreased, whereas the level of gain increased; respondents did not use escape strategies to cope.

## 4. Discussion

The obtained results indicate that the quality of life in the study group did not change significantly during the pandemic. Our analysis of the global sense of the quality-of-life correlates in the two cross-sectional surveys demonstrates certain regularities. The constant factors connected with the quality of life for people using psychological help during the pandemic are high personal gains and minor resource loss. This correlation remains in line with Hobfoll’s theory of conservation of resources [16]. In the first phase of the pandemic, active strategies were not linked to the sense of the quality of life. Similar results were obtained in a study of 146 high school students in the USA. Active coping was not associated with perceived stress and anxiety levels related to COVID-19 [34]. In a situation of social and physical limitations, the importance of active coping strategies directly related to self-efficacy decreases [35]. The quality of life, however, was diminished by several negative strategies, such as denial, venting, substance use, restraint, and self-blame; this pattern is characteristic of a situation of chaos or crisis. Research reports that people’s mental health deteriorated in the first period of the pandemic, characterized by increased anxiety, stress, and depression and decreased well-being and sleep quality [36]. After 3 months of the ongoing pandemic, the perceived quality of life was positively correlated with strategies related to seeking instrumental support and active coping. These strategies were not associated with the quality of life experienced during the pandemic onset; rather, they were activated while the pandemic was ongoing. After 3 months, strategies such as denying the existing situation and negative venting were not strongly linked to perceived quality of life. The personal experience of COVID-19 made the pandemic situation more real; in this case, these strategies became less relevant. In the same way, negative venting, which is more characteristic of the chaos phase, lost some of its relevance when the person adapted to the difficult situation; at this stage, the significance of strategies associated with being active and seeking instrumental support grew. This most likely points to the process of adapting to a difficult situation. Research conducted in Italy in November 2020 showed that the most common remedial strategies used by the medical service were: activity, planning, and acceptance [37].

In contrast, results of the longitudinal survey demonstrate that quality of life did not change despite the increased levels of factors negatively correlated with it. For all of the difficulties, as the pandemic wore on, the respondents were better off using a so-called helplessness strategy: they used psychoactive substances, refrained from taking action, and resorted to self-blame [38]. Another risk factor in the context of mental health was the decreased personal gains of the respondents. Research on the quality of life in a representative sample of Polish youth showed that during the COVID-19 pandemic, quality of life depended on such subjective resources as resilience, coping with stress, and ways of spending free time. The perceived quality of life increased with increased resilience, active coping methods, and free time for activities without using electronic media. It decreased with an increase in the frequency of using helplessness-like coping strategies [39]. Spanish research conducted during the COVID-19 pandemic showed that commitment coping was associated with positive adaptation, whereas coping without commitment correlated with a higher level of behavioural and emotional difficulties [40]. Similar relationships were verified in adult studies conducted in Poland: Active coping positively correlated with the perceived quality of life during the pandemic; helplessness-like coping showed negative associations with the perceived quality of life. Helplessness coping was also a significant mediator of the relationship between the distribution of resources and the perceived quality of life; it lowered the positive relationships between resource gains and perceived well-being and strengthened the negative relationships between resource loss and the perceived quality of life [38].

Our attempt at explaining which factors determine the quality of life after 3 months of the ongoing pandemic showed that the crucial factors are: a sense of quality of life before the occurrence of the pandemic, changes in the distribution of resilience resources, changed frequency of using escape strategies, and engaged coping; this became apparent, however, only in the second survey, when the chaos of adapting to a difficult situation was over. A study of dentistry students in Bangladesh showed a relationship between positive emotions and higher resilience and between lower levels of positive emotions and current satisfaction with social life [41]. Similarly, a cross-sectional study of Australian health care workers found that the most commonly used strategy positively related to health was staying active through exercise. On the other hand, the strategy associated with lowering mental well-being was the use of alcohol [42].

Given all our results, we can tentatively conclude that for the group using psychological help, two processes were increasingly operative during the pandemic. The first is associated with becoming adapted to a difficult situation, as illustrated by the resource gains, the emergence of active coping strategies after several months of the pandemic, and the less frequent use of escape strategies. The second is the development of a pandemic acute stress disorder, as illustrated by the results of the longitudinal survey, marked by decreased resource gains and the less frequent use of escape strategies.

## 5. Conclusions

Based on the results we obtained, it can be concluded that a high initial sense of QoL, high resource gain, and low resource loss add to one’s psychological capital, which plays an important role in functioning during a pandemic which presents long-term stress. According to Stevan Hobfoll, people who have numerous resources lose them more slowly; they can, even in a very difficult situation, activate processes of gaining new resources [13,43,44], and the outcomes of our research are in line with this conception. People who are high in resources are more likely to employ active coping strategies [27,43,45,46,47]. These strategies are also defined as engagement, because in addition to active coping they involve planning, reframing, or acceptance of the situation. Numerous authors have demonstrated that strategies involving activity did not play an adaptive role at the beginning of the pandemic [38,48,49]. In the face of an unknown threat and numerous restrictions imposed by external agencies, individual active coping did not produce a positive effect. However, after a period of adaptation to the pandemic situation, implementation of active, engaged coping entailed an adaptation process which enhanced perceived QoL. The risk factors for deterioration of mental health were, in turn, low initial sense of QoL and the use of escape-oriented stress coping strategies.

### 5.1. Limitations and Strengths

Our research involved a small sample that falls short of being representative, so extrapolating the results to a whole population is not appropriate. Additionally, there was a significant disproportion in terms of gender which made it impossible to demonstrate the differences between women and men. Nevertheless, the research showed some regularities which should be confirmed in further studies.

Regarding the future perspective, an examination of resource distribution and stress coping should be conducted with a representative sample from a specific, larger population in order to draw general conclusions. A resource such as sense of coherence could be explored, since it is directly linked to perceived stress and health condition.

On the other hand, this research has strengths. The most important of them is capturing the phenomenon of adaptation to the pandemic situation in Poland. Additionally, longitudinal studies distinguished the variables most closely related to the quality of life during the different periods of the pandemic.

### 5.2. Practical Implications

The practical implications that can be inferred from the present research concern:The need for continuous health monitoring;Prudent implementation and prudent relaxation of regulations restricting social activity in a way that while protecting individuals from SARS-CoV-2 does not cause severe quality of life deficits in persons using psychological help; andPlanning and implementing mental health support in parallel with restrictions imposed during the pandemic.

Our research findings demonstrate the significant role of psychological capital as a factor protecting mental health. Therefore, while taking preventive and therapeutic action, one needs to focus on enhancing the process of acquiring resources; however, to be able to acquire them, such resources must be available locally, including among others, psychological and psychiatric help, the possibility of establishing interpersonal relations, and opportunities to satisfy developmental needs (learning, work, interests, etc.). Based on the present research, special emphasis should be placed on the development of destructive coping strategies forming the so-called helplessness triad (psychoactive substance use, restraint, self-blame), which can trigger the emergence of such psychopathological syndromes as depressive addiction disorder and pandemic acute stress disorder.

## Figures and Tables

**Table 1 ijerph-19-06023-t001:** Results for Quality of Life, Stress Coping Strategies, and Resource Gain and Loss in Two Pandemic Periods: Longitudinal Study Results.

Variables	Wave 1 ^a^		Wave 2 ^b^		Test of Differences			Test of Correlations	
	*M*	*SD*	*M*	*SD*	t(82)	*p*	Cohen’s *d*	*r*	*p*
Quality of life:									
Physical health	14.19	2.43	13.97	2.39	0.84	0.41	0.09	0.51	0.01
Psychological field	14.17	2.85	13.91	2.84	0.79	0.43	0.09	0.46	0.01
Social relations	14.31	3.25	14.18	3.33	0.36	0.72	0.04	0.52	0.01
Environment	13.98	2.23	14.08	2.38	−0.41	0.69	−0.04	0.44	0.01
Global quality of life	14.16	2.21	14.04	2.26	0.51	0.61	0.06	0.50	0.01
Coping strategies:									
Active coping	1.70	0.76	1.61	0.78	0.86	0.39	0.09	0.23	0.04
Planning	1.87	0.77	1.81	0.71	0.50	0.62	0.06	0.11	0.32
Positive reframing	1.72	0.87	1.81	0.78	−0.86	0.40	−0.09	0.32	0.01
Acceptance	1.89	0.69	1.89	0.71	−0.07	0.95	−0.01	0.32	0.01
Humour	0.90	0.51	1.01	0.67	−1.35	0.18	−0.15	0.33	0.01
Religion	1.30	1.16	1.19	1.09	0.91	0.37	0.10	0.53	0.01
Seeking emotional support	1.95	0.79	1.92	0.80	0.33	0.74	0.04	0.20	0.06
Seeking instrumental support	1.85	0.78	1.76	0.79	0.93	0.36	0.10	0.36	0.01
Self-distraction	1.81	0.84	1.66	0.76	1.46	0.15	0.16	0.26	0.02
Denial	0.57	0.70	.59	0.69	−0.25	0.81	−0.03	0.17	0.13
Venting	1.42	0.66	1.39	0.77	0.35	0.73	0.04	0.41	<0.001
Psychoactive substance use	0.24	0.51	0.42	0.65	−2.22	0.03	–0.24	0.26	0.02
Restraint	0.56	0.56	0.78	0.68	−2.68	0.01	−0.29	0.31	0.01
Self-blame	0.66	0.78	0.92	0.80	−2.17	0.03	−0.24	0.01	0.93
Resources:									
Management resources (gain)	2.38	0.87	2.09	0.88	3.20	<0.001	0.35	0.54	0.01
Social status resources (gain)	2.00	1.00	1.74	1.04	2.22	0.03	0.24	0.48	0.01
Resilience resources (gain)	2.59	0.86	2.34	0.88	2.96	0.00	0.32	0.62	0.01
Family resources (gain)	1.09	1.09	0.97	1.05	1.04	0.30	0.11	0.52	0.01
Material status resources (gain)	2.07	0.80	1.93	0.97	1.54	0.13	0.17	0.57	0.01
Growth resources (gain)	1.61	1.11	1.41	1.07	1.58	0.12	0.17	0.44	0.01
Community resources (gain)	2.85	1.03	2.46	1.02	3.66	<0.001	0.40	0.55	0.01
Resourcefulness (gain)	2.16	0.77	1.91	0.81	3.14	<0.001	0.34	0.58	0.01
Management resources (loss)	0.77	0.84	0.97	0.99	−1.67	0.10	−0.18	0.31	0.01
Social status resources (loss)	0.92	1.05	1.01	1.08	−0.72	0.48	−0.08	0.47	0.01
Resilience resources (loss)	0.90	0.83	1.06	0.91	−1.33	0.19	−0.15	0.22	0.04
Family resources (loss)	0.36	0.66	0.54	0.93	−1.70	0.09	−0.19	0.33	0.01
Material status resources (loss)	0.38	0.57	0.52	0.96	−1.21	0.23	−0.13	0.21	0.06
Growth resources (loss)	0.73	0.88	0.91	0.97	−1.69	0.10	−0.18	0.45	0.01
Community resources (loss)	0.74	0.83	0.82	0.85	−0.66	0.51	−0.07	0.23	0.03
Resourcefulness (loss)	0.75	0.73	0.88	0.85	−1.34	0.19	−0.15	0.36	0.01

*Note*: ^a^ first wave result, ^b^ second wave result.

**Table 2 ijerph-19-06023-t002:** Results of Hierarchical Multiple Regression Analysis for the Second QoL Measurement.

Variables	Step 1 β	Step 2 β	Step 3 β
Quality of Life 1 ^a^	0.50 ***	0.62 ***	0.38 ***
Resilience resources (loss) 1		0.23 ^	−0.14
Resilience resources (gain) 1		−0.08	0.16
Escape coping style		0.07	0.04
Engaged coping style 1		0.27 *	0.10
Resilience resources (loss) 2 ^b^–Resilience resources (loss) 1			−0.36 ***
Resilience resources (gain) 2–Resilience resources (gain) 1			0.33 ***
Escape coping style 2–Escape coping style 1			−0.21 *
Engaged coping style 2–Engaged coping style 1			−0.07
R^2^	0.24	0.30	0.54
R^2^ change	0.25 ***	0.09 *	0.24 ***

*Note*: ^a^ first wave result, ^b^ second wave result. *** *p* < 0.001, * *p* < 0.05, ^ *p* < 0.1.

## Data Availability

All data generated or analysed during this study are available on request to authors.

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
