# Peer review of "Determinants of Quality of Life in the COVID-19 Pandemic Situation among Persons Using Psychological Help at Various Stages of the Pandemic"

_ijerph, 2022, doi:10.3390/ijerph19106023_

Round 1

Reviewer 1 Report

The study is very interesting but needs to be supported by more bibliographic references on this subject.
Regarding the ethics committee, it would be necessary for the authors to specify if they requested it and if they did not, at least clarify if the patients signed a declaration of participation.

Line 48: “The psychological consequences of living in  the pandemic have a varying intensity but they affect entire populations”

It is necessary to explain more this idea and to add more references because in the last year there are a lot of studies that refers psychological problems in pandemic.

Line 71, 72

“During the COVID-19 pandemic these constructs have been impaired, both due to the disease itself, fear of it and all of the restrictions imposed on  community life and ways of working”.

It is important to details more this concept and add references that support it.

Line 82 (Götmann & Bechtoldt,  2021)

The reference is wrong in APA style, correct it.

Line 108,109: “The perceived QoL is a measure of a person's  adaptation to a specific situation; it is influenced by many variables, most importantly by  the processes of losing and gaining resources and ways of coping in difficult situations”.

It is important to explain the variables and support it with references about the instrument.

2.1. Subjects and Procedure

It is no clear why the participants were recruited in different moments and ages. Please, justify it.

Another question ¿It was administer the same survey?,

2.2. Line 186 etics

“The research was conducted in accordance with the ethical standards of a responsible committee on human experimentation (institutional or regional)”

Please, clarify if the research was approved by a committee or no. If you have a code o date of register, please, introduce it.

Line 190 : “The participants were informed about the confidentiality and anonymity of the research, and that they had the right not  to participate in the survey”.

It is important the way of inform, because people were confined, it is important explain if the survey has a part with this or the research inform by other way.

Line 240: “The method enables an assess-240 ment of how often an individual uses 14 different coping strategies”

It is necessary explain the 14 different coping strategies.

Line 245: In this study, the instructions for the respondents were slightly modified to adapt to coping with the COVID-19 situation

If you modified the instrument ¿how affect to validity and reliability?

Line 365: Discussion, line 403 conclusions

In general these parts  must be improved and developed  with more studies.

Author Response

Dear Sir/Madam,

Thank you very much for reading our article and for all your suggestions for improving it. Below we present information about modifications.

  • The study is very interesting but needs to be supported by more bibliographic references on this subject.

Thank you for your favorable evaluation of our work. Additional references have been added.

  • Regarding the ethics committee, it would be necessary for the authors to specify if they requested it and if they did not, at least clarify if the patients signed a declaration of participation.

The research was conducted according to the standards of good research practice recommended by the American Psychological Association. The participants were informed about the confidentiality and anonymity of the research and that they had the right not to participate in the survey. Additionally, the very first question on the survey was: Do you agree to participate in the study voluntarily? The possible answers were Yes and No. Finally, any personal data were collected.

  • Line 48: "The psychological consequences of living in the pandemic have a varying intensity but they affect entire populations" It is necessary to explain more this idea and to add more references because in the last year there are a lot of studies that refers psychological problems in pandemic.

Additional references have been added.

  • Line 71, 72

"During the COVID-19 pandemic these constructs have been impaired, both due to the disease itself, fear of it and all of the restrictions imposed on  community life and ways of working". It is important to details more this concept and add references that support it.

Additional references have been added.

  • Line 82 (Götmann & Bechtoldt, 2021) The reference is wrong in APA style, correct it.

The citation method of this item has been modified according to APA style.

  • Line 108,109: "The perceived QoL is a measure of a person's adaptation to a specific situation; it is influenced by many variables, most importantly by  the processes of losing and gaining resources and ways of coping in difficult situations". It is important to explain the variables and support it with references about the instrument.

Thank you for this suggestion. We decided to delete this sentence because another reviewer also noted this and suggested that this sentence adds nothing to the research problem.   

  • It is no clear why the participants were recruited in different moments and ages. Please, justify it.

The research was conducted at different moments of pandemic because we wanted to capture the process of adaptation to a difficult situation in the first phase and after passing the initial impact. Therefore, we addes this sentence to the text.

  • Another question ¿It was administer the same survey?,

This study aimed to compare the process of adaptation to difficult situations during different moments of the pandemic. So we use the same questionnaire in all studies. In addition, we have added a corresponding explanation in the text.

  • "The research was conducted in accordance with the ethical standards of a responsible committee on human experimentation (institutional or regional)" Please, clarify if the research was approved by a committee or no. If you have a code o date of register, please, introduce it.

We did not apply for consent to the examination to the ethics committee. However, the research was conducted following the standards of the American Psychological Association. Additionally, we did not collect information on intimate matters.

  • Line 190 : "The participants were informed about the confidentiality and anonymity of the research, and that they had the right not to participate in the survey". It is important the way of inform, because people were confined, it is important explain if the survey has a part with this or the research inform by other way.

Before the research started, we asked the first question: Do you agree to participate in the study voluntarily? The possible answers were Yes and No. After that, any personal data were collected.

  • Line 240: "The method enables an assess-240 ment of how often an individual uses 14 different coping strategies" It is necessary explain the 14 different coping strategies.

Thank you very much for this suggestion. We add information about the names of 14 different coping strategies in the text.

  • Line 245: In this study, the instructions for the respondents were slightly modified to adapt to coping with the COVID-19 situation. If you modified the instrument ¿how affect to validity and reliability?

Thank you very much for this suggestion. Reliability was calculated and compared to the original version of the method. The results were similar. We present reliability (alfa) in the text.

  • Line 365: Discussion, line 403 conclusions. In general these parts must be improved and developed  with more studies.

Your suggestion has been included.

Reviewer 2 Report

In this work, authors analyze the determinants of quality of life in the COVID-19 Pandemic among persons using psychological help through cross sectional and longitudinal studies. Numerous factors were assessed, although modest sample size is presented and the previous psychological status of the participants are not considered in these analysis. I find this paper interesting due to provide relevant information due to situation we have lived. However, some suggestions are proposed to improve the work.

Abstract:

  • I do not understand the following sentence “The results of our longitudinal survey demonstrate that the severity of depressive symptoms increased during the pandemic.” (line 22).

Introduction section:

  • The introduction is exhaustive and provides several studies regarding the topic in general, but it may be too long, specially section 1.2
  • I consider the following sentence does not add much information “The perceived QoL is a measure…” (line 108).

Materials and methods section:

  • More information about the procedure is necessary: There were any inclusion and exclusion criteria? what kind of psychological help had used the participants? How did you have access to the participants? How long did participants take to fill out the questionnaire?
  • I am never convinced by providing results in the participant’s section (line 161-167; line 168-174; line 179-184). It would help the reader if you report it at result section.
  • I think that the 2.2 Ethics section is incomplete.

Results:

  • There is too much information in table S1, as well as, some variables are in polish. It is very difficult to understand it.
  • Footnotes from table 1 are missing
  • Just a reminder: Correlation does not imply causation. It is important to be taken into account when writing the results and explain the main conclusions.

Discussion:

  • What about the psychopathological status from the participants? Do you think that could have influenced the results? And the individual’s differences, such as gender? It is well known that there are differences between males and females when seeking help and even to using different coping strategies. It would be interesting for the reader if you consider to discuss it in the discussion section.
  • What are the strengths of your study? Please, consider to include it.
  • In 5.2 Practical implication you mention “severe mental health”, “psychological capital as a factor protecting mental health”, but your analyses are about QoL and coping strategies. Perhaps it would not be appropriate to extrapolate at this level (just a suggestion…)

Minor comments:

  • There are format differences when reporting some references along the text (eg. Line 82).
  • Remember to name the full word first and then the abbreviation (eg. UN)

Author Response

Dear Sir/Madame,

Thank you very much for reading our article and for all your suggestions for improving it. Below we present information about modifications.

  • Abstract: I do not understand the following sentence "The results of our longitudinal survey demonstrate that the severity of depressive symptoms increased during the pandemic." (line 22).

Thank you very much for this suggestion. We modified these sentences into "The results of our longitudinal surveys demonstrate that escapism increase." 

Introduction section:

  • The introduction is exhaustive and provides several studies regarding the topic in general, but it may be too long, specially section 1.2

Thank you for this suggestion. In our opinion, the specificity pandemic context in Poland is important for understanding our results.

  • I consider the following sentence does not add much information "The perceived QoL is a measure…" (line 108).

Thank you for this suggestion. We decided to delete this sentence because another reviewer also noted this and suggested that this sentence is not clear and adds nothing to the research problem.

  • Materials and methods section: More information about the procedure is necessary: There were any inclusion and exclusion criteria? what kind of psychological help had used the participants? How did you have access to the participants? How long did participants take to fill out the questionnaire?

Thank you for this suggestion. We added detailed information about the procedure in the text: "Additional inclusion criteria were: age over 18 years, included in the register of clients of counseling center run by NGO. The surveys were conducted remotely via the Google platform. The invitation for the research was sent via e-mail.”

AND

In all three studies, participants administrate the same questionnaire. Filling up the survey took about 20 minutes. There were different subjects taken in each study. The research was conducted at different moments of the pandemic because we wanted to capture the process of adaptation to a difficult situation in the very first phase and after passing the initial impact.

  • I am never convinced by providing results in the participant's section (line 161-167; line 168-174; line 179-184). It would help the reader if you report it at result section.

Thank you for these suggestions. We moved the sample description to specific paragraphs in which we present a specific study. 

  • I think that the 2.2 Ethics section is incomplete.

We have supplemented this section with additional information regarding i.a., informed consent.

  • Results: There is too much information in table S1, as well as, some variables are in Polish. It is very difficult to understand it.

Thank you for this suggestion. We changed the names of dimensions to English. Much information is presented in the table, but we think they are necessary in this text to present the results better.

  • Footnotes from table 1 are missing.

The footnote in the table has been added. 

  • Just a reminder: Correlation does not imply causation. It is important to be taken into account when writing the results and explain the main conclusions.

Thank you very much for this reminder. We are aware of this. We tried to present our result in terms of a correlational approach. We modified this pattern in the description of regression results in longitudinal studies

  • Discussion: What about the psychopathological status from the participants? Do you think that could have influenced the results? And the individual's differences, such as gender? It is well known that there are differences between males and females when seeking help and even to using different coping strategies. It would be interesting for the reader if you consider to discuss it in the discussion section.

Thank you for this suggestion. We are aware of possible gender differences in coping styles and seeking psychological help. Unfortunately, there was a significant disproportion in our sample in terms of gender, which made it impossible to make meaningful comparisons. We have added a note about this in the limitations section.

  • What are the strengths of your study? Please, consider to include it.

Thank you for this suggestion. We added to the text strengths of our research

  • In 5.2 Practical implication you mention "severe mental health", "psychological capital as a factor protecting mental health", but your analyses are about QoL and coping strategies. Perhaps it would not be appropriate to extrapolate at this level (just a suggestion…)

Thank you for this suggestion. We modified it.

  • Minor comments:

There are format differences when reporting some references along the text (eg. Line 82).

Remember to name the full word first and then the abbreviation (eg. UN)

Thank you for this suggestion. We spelled abbreviations of COVID and UN in the text.

Reviewer 3 Report

Thank you for the opportunity to review this paper. The covid-19 pandemic is an emerging, rapidly evolving situation, and rigorous information is needed. Here I provide some suggestions that may help to improve the quality of this work:

Abstract: I recommend including the mean age of the participants and the percentage of predominant gender for each survey.

Introduction:

Page 2, line 82. The citation of (Götmann & Bechtoldt, 2021) should be numbered, and all subsequent number citations should be numbered again.

Page 2, line 87. The citation of (Dawson & Golijani-Moghaddam, 2020) should be numbered, and all subsequent number citations should be numbered again.

Material and methods:

Subjects and procedure:

-Could you please provide the inclusion and exclusion criteria?

-How was the recruitment done?

-Some participants are the same in study 1 and study 2?

-Since the two cross-sectional studies involved persons who in the 12 months preceding the outbreak of the pandemic had used psychological help at least once, do you know what kind of disorders did the have? These disorders could affect the way people cope with difficult situations (like the pandemic situation)?

Ethics:

-Was there a written informed consent?

Results:

-Consider a Table for the sociodemographic charactheristics of the samples.

-Consider providing the QoL outcome values in each study.

Discussion: 

-The Discussion section should be started remembering the aim of the study.

-Consider the fact that in in March 2020 most of the people was afraid to the covid-19, rules imposed by governments to avoid spreading the virus were clear and many people was committed to following the rules, creating a feeling of community, common well and social support that could act as coping and protection factor. Instead, by July 2020 restrictions were relaxed, many people stopped following the existing rules, fake news had increased, individualism reigned, and people who followed the rules had to deal with those who did not (and put their health and that of the community at risk).

-The fact that the sample was clinical sample (persons who in the 12 months preceding the outbreak of the pandemic had used psychological help at least once) should be pointed in the Limitations section. Coping strategies could be different from the general population.

-Page 10, lines 431-433. Please change “prudent implementation of regulations restricting social activity” for “prudent implementation and prudent relaxation of regulations restricting social activity”. Results pointed the negative coping. Please change “deficits in the whole society” to “deficits in persons using psychological help”. (Your sample was persons using psychological help, not general population, so you cannot state that).

Author Response

Dear Sir/Madame,

Thank you very much for reading our article and for all your suggestions for improving it. Below we present information about modifications.

  • Abstract:I recommend including the mean age of the participants and the percentage of predominant gender for each survey.

Thank you very much. The suggestion was included. We added M and SD to the abstract.  

  • Introduction:

Page 2, line 82. The citation of (Götmann & Bechtoldt, 2021) should be numbered, and all subsequent number citations should be numbered again.

Page 2, line 87. The citation of (Dawson & Golijani-Moghaddam, 2020) should be numbered, and all subsequent number citations should be numbered again.

 Thank you very much for pointing this. Your suggestion was included

  • Material and methods:

Subjects and procedure:

-Could you please provide the inclusion and exclusion criteria?

-How was the recruitment done?

-Some participants are the same in study 1 and study 2?

-Since the two cross-sectional studies involved persons who in the 12 months preceding the outbreak of the pandemic had used psychological help at least once, do you know what kind of disorders did the have? These disorders could affect the way people cope with difficult situations (like the pandemic situation)?

Thank you very much for this suggestion. We added this information (inclusion and exclusion criteria; recruitment) in the section with the procedure description. There were not the same participants in studies 1, 2, and 3. We clarified this information in the text.

There were no subjects with psychopathological disorders diagnosis in our sample, so the risk of modifying the results by their psychopathology is low. Furthermore, our sample included only people with mild psychological problems.        

4) Ethics:

-Was there a written informed consent?

There was no written informed consent, but we asked the first question before the research started: Do you agree to participate in the study voluntarily? The possible answers were Yes and No. After that, any personal data were collected.

4) Results:

-Consider a Table for the sociodemographic characteristics of the samples.

-Consider providing the QoL outcome values in each study.

Information on the general level of quality of life is presented in a table with correlations and a table presenting a comparison of measurements in longitudinal studies. Sociodemographic characteristics of the samples are presented only in the text because of  the length of the article

5) Discussion: 

-The Discussion section should be started remembering the aim of the study.

Thank you very much for this suggestion. We decided not to put the aim of the study at the beginning of the discussion, but we presented our results according to these goals.

 -Consider the fact that in in March 2020 most of the people was afraid to the covid-19, rules imposed by governments to avoid spreading the virus were clear and many people was committed to following the rules, creating a feeling of community, common well and social support that could act as coping and protection factor. Instead, by July 2020 restrictions were relaxed, many people stopped following the existing rules, fake news had increased, individualism reigned, and people who followed the rules had to deal with those who did not (and put their health and that of the community at risk).

Thank you for these suggestions. We agree that the power of restrictions may influence functioning people. Perhaps it is related to different stress coping and more active strategies. Again, thanks to your suggestion, we added this information to the discussion.

 -The fact that the sample was clinical sample (persons who in the 12 months preceding the outbreak of the pandemic had used psychological help at least once) should be pointed in the Limitations section. Coping strategies could be different from the general population.

Thank you very much for this suggestion. We are aware of it and pointed out such limitations in the first sentences of the section with limitations.

-Page 10, lines 431-433. Please change "prudent implementation of regulations restricting social activity" for "prudent implementation and prudent relaxation of regulations restricting social activity". Results pointed the negative coping. Please change "deficits in the whole society" to "deficits in persons using psychological help". (Your sample was persons using psychological help, not general population, so you cannot state that).

Thank you very much for this suggestion. We modified these two sentences in the text.

Reviewer 4 Report

Please spell abbreviations when you use them first time, for example, Line 33 UN, Line 34 COVID-19.

Good luck

Author Response

Dear Sir/Madame,

Thank you very much for reading our article and for all suggestions for improving it. Below we present information about modifications.

We spelled the abbreviation of COVID and UN in the text. Thank you for your wishes.